# GRAPH PRIORS FOR DEEP NEURAL NETWORKS

**Francis Dutil**[*] **& Joseph Paul Cohen**[*] **& Martin Weiss & Yoshua Bengio**
Montreal Institute for Learning Algorithms
Université of Montréal
{dutilf,cohenjos,weissmar,bengioy}@iro.umontreal.ca

**Georgy Derevyanko**
Department of Chemistry and Biochemistry
Centre for Research in Molecular Modeling (CERMM)
Concordia University
georgy.derevyanko@gmail.com

## ABSTRACT

In this work we explore how gene-gene interaction graphs can be used as a prior for the representation of a model to construct features based on known interactions between genes. Most existing machine learning work on graphs focuses on building models when data is confined to a graph structure. In this work we focus on using the information from a graph to build better representations in our models. We use the percolate task, determining if a path exists across a grid for a set of node values, as a proxy for gene pathways. We create variants of the percolate task to explore where existing methods fail. We test the limits of existing methods in order to determine what can be improved when applying these methods to a real task. This leads us to propose new methods based on Graph Convolutional Networks (GCN) that use pooling and dropout to deal with noise in the graph prior.

## 1 INTRODUCTION

Gene expression data produced from RNA-Seq or MicroArray is typically thought of as many arbitrary variables without taking advantage of observed relationships between genes. As researchers piece together how they relate to each other, gene network graphs are being built from the papers they publish. Genes are said to interact with each other if they (or the protein they code for) have a physical or a functional association. We can also consider correlations mined from large gene expression datasets implying unknown relationships.

We explore how these graphs can be used as priors influencing the representation learned by a model in order to guide learning toward more relevant features. This can help the model to ignore noise which correlates with a target prediction by chance, which happens a lot in datasets where there are more input variables than training examples. If done correctly, we can also reduce the overall number of parameters, analogous to what can be done with a Convolutional Neural Network.

In order to build a better understanding of these complex biological systems and in order determine how existing methods can be improved we model the system as a percolation task [Broadbent & Hammersley (1957)] representing a biological pathway. The percolation task is: given a graph with a set of source nodes, sink nodes, and intermediate nodes, with input values $\{0, 1\}$ for each node does there exist a path from a source to a sink. If so this graph is said to percolate. We modify this task by changing the graph data in three ways to simulate problems in real datasets as shown in Figure 1. We can add extra uninformative unconnected nodes which are not part of the graph and not needed to solve the percolation task. We can also add extra uninformative connected nodes which are present in the graph and act as incorrect gene relationships or relationships that are unrelated to the task. We can also degrade the percolation graph itself by removing edges to simulate not-yet-known interactions.

## 2 GRAPH PRIORS

Most existing work focuses on building models when data is confined to a graph structure. In this work we focus on using the information from a graph (or graphs) to build better representations in

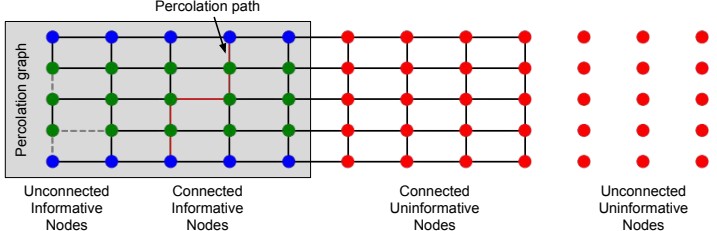

Figure 1: We model how gene-gene interaction graphs can overlap real data using the percolate task as a proxy for a pathway between genes. Blue nodes represent the source and sink of which signal must propagate through Green nodes to percolate. Red nodes are unrelated to the problem and may or may not have edges in the graph which would be a distraction to the main problem.

our models. When working with point clouds, social networks, or protein structures the graph is fundamental and unchangeable. With gene expression information the graphs are complementary to the main task and can act as a prior. With low numbers of samples, a common setting in biology, relationships between variables can provide signal to help a model avoid learning erroneous correlations.

**Via regularization:** The method proposed by Min et al. (2016) is to regularize the weights of a Sparse Logistic Regression (SLR) based on the connectivity of the nodes found in the interaction graph. This is achieved by adding a regularization term $\lambda |w|^T L |w|$, with the graph Laplacian $L$ to a logistic regression loss. This regularization encourages the weights to associate with nodes that have a high number of interactions to remain important.

**Via convolution:** We can also use the structure of the graph as a prior. By performing convolution operations on a node to incorporate its neighbors we can extract and propagate the features along the edges of the graph, like what happens inside a Convolution Neural Network with adjacent pixels. This convolution over the features $\theta * X$ is not trivial when the structure of the graph is highly complex. In Bruna et al. (2014) they explore the use of highly sparse MLPs where each feature is only linked to its neighbours. Bruna et al. (2014) also used spectral convolution, by projecting the parameters $\theta$ into the spectral space of the Laplacian matrix $L$: $X^{t+1} = \theta * X^t = U diag(\theta) U^T X^t$, where $U$ contains the eigenvectors of the Laplacian $L$.

However, the full projection of the eigenvectors represents paths of infinite length and will therefore take into account all nodes at once and prevent the network from reasoning about interactions with neighbors. This mean that no locality is present in the convolution, which makes the interpretability and the sharing of the parameters a lot more difficult. To obtain this locality in the convolution, we can utilize methods in Defferrard et al. (2016) and Kipf & Welling (2016) and approximate the convolution to the first neighbouring layers (paths of length 1) for each nodes. With $A' = A + I_N$ and $D'_i = \sum_j A'_{ij}$ this leave us with:

$$X^{t+1} = diag(\theta) * X^t \approx D'^{-1/2} A' D'^{-1/2} X^t \theta$$

To increase the receptive field of each nodes, we can then simply add convolution layers on top of each other. This approach however doesn't allow us to have different types of interactions (all nodes are aggregated before any transformation is done). While it is possible to have different sets of parameters for different interactions like in Bruna et al. (2014), not all graphs (specifically gene interaction graphs) have different types of edges for each type of interaction between genes. In Duvenaud et al. (2015) they deal with this problem by considering the degree of each node as different kinds of interaction. However, in the case of genes, the distribution of the degree of the nodes varies too much for this trick to be practical. In our case, we have done like in Hamilton et al. (2017) and added a skip connection at each convolution layer, which essentially preserves two kinds of signals: the neighbourhood and the node itself. We can utilize this method to combine multiple graph priors in the same network. The full convolution can then be constructed in layers with an activation function and aggregation clustering method:

$$X^{t+1} = Aggregate(ReLU(\tilde{A}X\theta_1 + X\theta_2))$$

In this paper, we have used hierarchical clustering based on the node connectivity in the interaction graph to reduce the number of nodes by 2 after each convolution. A max pooling is then done on each resulting clusters.

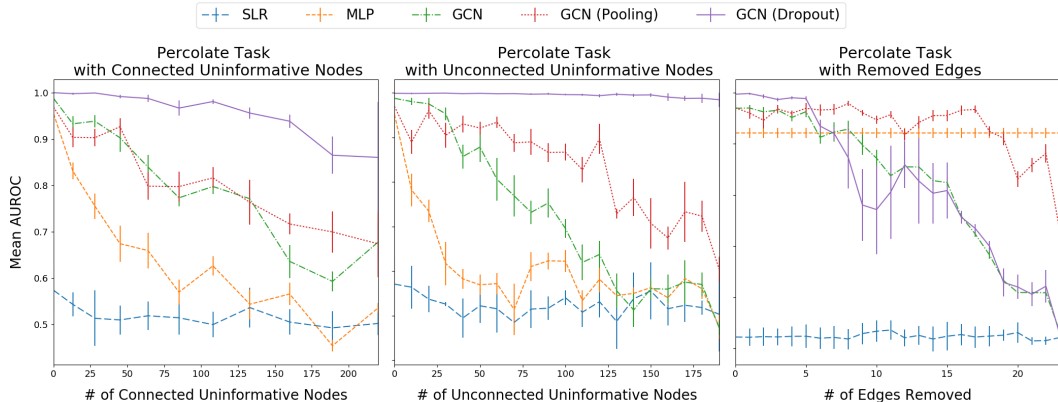

Figure 2: Here we vary the two types of extra nodes and how many edges are removed beyond a base 4x4 percolate graph. We generate 1000 samples split into train/valid/test split with 60%/20%/20% proportions and present the average test AUC over 10 runs.

| Dataset | Random | SLR | MLP | GCN | GCN (with pooling) | GCN (with dropout) |
|---------|--------|-----|-----|-----|--------------------|--------------------|
| GBM-Survival | 51.3%±3.8 | 64.5%±1.5 | 60.9%±5.1 | 61.5%±2.4 | 66.6%±2.2 | 64.0%±1.7 |
| Percolate-Equiv | 47.5%±1.5 | 54.3%±3.2 | 51.0%±1.6 | 51.2%±3.6 | 52.8%±2.0 | 51.2%±2.3 |

Table 1: Here we evaluate these models on large real and synthetic datasets. (AUC over 10 runs). Currently these numbers do not show a significant increase in performance but our analysis at a smaller scale indicates that we are closer to a method that may one day work.

To help with the low amount of data, we also experiment with Drop-out Srivastava et al. (2014). After each convolution layer, each node has a 40% change of being dropped. The model can't then rely on some specific node and has to pass the important information across the network, which in turn can make the learning of important features easier.

## 3 EXPERIMENTS

To explore existing methods in Figure 2 we utilize our modified percolate task which differentiates existing methods well when varying the extra nodes or missing edges. We explore the limits of these methods in order to provide signal on what can be improved when applying to a real task. We find that the MLP model is not capable of handling extra connected or unconnected nodes and performance drops immediately while the GCN models are able to better deal with this noise. We find that utilizing dropout with the GCN yields almost no performance drop unless informative edges are removed. We interpret this as when nodes are connected they can pass information through their neighbors and avoid being impacted by dropped out nodes. We also observe that pooling helps when edges are missing which could be attributed to the imputation of missing edges. The extra connected nodes and disconnected edges explore how important the quality of the graph is because noise in the graph removes the effect of the prior. These results suggest that high quality biological interaction graphs should be used as prior to help any given task.

In Table 1 we evaluate these methods on a prototypical biological dataset used by Min et al. (2016) composed of gene expression profiles for 440 glioblastoma (GBM) patients in the TCGA database [The Cancer Genome Atlas et al. (2013)] and a Protein-Protein Interaction (PPI) graph from Pathway Commons [Cerami et al. (2011)]. The task is to predict 440 patient's survival within one year. We construct a Percolate-Equiv dataset which is a version of the percolate task at a scale to match the GBM problem. We consider a 6x6 percolate graph as the core pathway to be learned and add 1260 uninformative connected and 1000 uninformative unconnected nodes as noise.

To conclude, we present a proxy task to study how to incorporate gene pathway networks into deep learning models. We gain insights into where the limitations are of the existing algorithms. We propose applying GCNs in order to utilize one or many gene networks and we find that using dropout maintains performance while other methods cannot handle the noise.

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
