# OpenReview forum: "Graph Priors for Deep Neural Networks"
_ICLR.cc/2018/Workshop — Reject_

### Official Review · AnonReviewer1 · 2018-03-03
**Still very preliminary**

**Rating:** 3
**Confidence:** 4

**Review:**

In this work, the authors propose to use known graph structures, such as gene-gene interaction networks, as a structure for guiding convolutions for learning embeddings.

The work borrows heavily from the literature, so it is not clear that there is any real methodological contribution in the current work. Second, the connection between the percolation task and the proposed approach is not clear; for example, the experimental section mentions that the “percolate graph [is] the core pathway to be learned.” However, the biological dataset simply uses the PPI network for convolution and (seems to) learn to predict patient outcomes, not pathways. The work does not draw any connections to a probabilistic interpretation of their model, so the use of the term “prior” is rather suspect.

Biologically, it is not apparent that the proposed method accounts for different types of gene-gene interactions (for example “up-regulates” compared to “down-regulates”). Similarly, it is not clear whether the convolutional approach in this work preserves any semblance of interpretability; thus, its utility in actual biomedical applications is not clear.

The results are not especially encouraging; indeed, the proposed approach is not statistically distinguishable from a standard MLP.

While the idea of using domain knowledge to improve learning is promising, I find this work too preliminary for presentation to a wider audience at this time.

---

### Official Review · AnonReviewer4 · 2018-03-08
**The topic is interesting, but the problem definition would be unclear**

**Rating:** 5
**Confidence:** 4

**Review:**

It is very interesting to investigate how the graph structure can be used as priors, and this paper fits well to the workshop track for late-breaking results. The paper discusses a specific case of 'percolation tasks' they defined, but the structural regularization/priors would be one of important auxiliary information in many real-world contexts including life sciences. Moreover, regularization terms of ML in the optimization formulation often corresponds to the priors in theBayesian settings, and thus the topic investigates a quite promising theme.


But the following points in the manuscript are still unclear and confusing. I understand we have the 3-page limitation, but it would be nice to improve these points.


1. Definition of the 'percolation task'. (for example, Figure 2)

How did you generate the training data for Figure 2?
Generate 1000 grid graphs based on 4x4 graph with random edge removals with some fixed probability, add two types of noisy nodes to perturb, and predict the sources-sinks reachability only from a given graph?

The minimum explanations on these unclearness would be nice to have. The word 'percolation' in the Broadbent-Hammersley's sense reminds us the setting includes some probabilities (open or closed bond with probability p), but the current description of the 'percolation task' would be ambiguous.

In addition, the reachability check is one of simple examples where, for example, Gated Graph Neural Networks (Li+ ICLR2016) can be solved (It was actually explained at Section 3 in the paper). But GGNN usually requires the edge types, and I wonder if there is any relationship between this 'reachability check' and 'parcorlation task'.... Note that I'm talking about GGNN, not GG 'Sequence' NN which can output sequences, and the main target of the paper above.


2. Prediction related to gene interactions (what is x and y of this supervised learning?)

The "graph Laplacian regularizers", as applied to logistic regression in Min+ 2016, is now widely used in many other contexts. But in these situations (including Min+, 2016), the target problem is a 'supervised learning', and we have the response y to be predicted, and some x as observable features, in addition to the graph-structure constraints on x such as gene-gene interactions or protein-protein interactions.

So, what is y and x in this 'percolation task' setting? x is a 0-1 value on each node (what do 0 and 1 mean?) and y is the source-sink reachability...? What insights can we have for gene-gene interactions if this can be solved in a supervised manner?


3. What are predicted from gene expression profiles?

Gene expression profiles would usually be real-valued, and gene-gene networks are not grid-like. But, the targeted 'parcolation task', are variables x binary, and what does 'Percolate-Equiv' (based on 6x6 graphs?) means? How to reduce 'Min+, 2016' dataset for Table 1 into binary problems? Is it different from the original prediction problem of their paper..?? This point was very confusing to me.


The list of pros and cons

[pros]
1. The target topic is very interesting
2. Some experimental evidences are provided to get some insights
3. It can be applied to real-world problems of life sciences

[cons]
1. The problem definition is quite ambiguous to understand the results
2. The relationship to their 'percolation task' and Min+ 2016's original problem is seemingly unclear
3. It is not clear whether the main focus is a general interest or a specific interest to prediction based on gene-gene interactions.

---

### Official Review · AnonReviewer3 · 2018-03-15
**Difficult to read**

**Rating:** 3
**Confidence:** 5

**Review:**

This workshop submission is hard to read. In more details:

1. It is almost impossible to understand what the submission really does from the abstract.
For instance, in the first sentence of the abstract, the author wrote: "we explore how gene-gene interaction graphs can be used as a prior for the representation of a model to construct features" ...
+ what model? why is this a significant task?

In the fourth sentence of the abstract, the author wrote: "We use the percolate task, determining if a path exists across a grid for a set of node values, as a proxy for gene pathways."
+  how does the percolate task relate to the task of representation learning?

2. The submission has many grammar mistakes, for instance:
- third paragraph: "in order determine"
- third paragraph:  "for each node does there exist a path from a source to a sink."
- section 2, first paragraph: missing citations in its first sentence.
- "this leave us with:"

3. Many math notations are used without a clear definition first, for instance:
- \theta is never defined but is used
- A is never defined but is used

4. The percolation task, why is it relevant?
- the author wrote: "In order to build a better understanding of these complex biological systems and in order determine how existing methods can be improved we model the system as a percolation task [Broadbent & Hammersley (1957)] representing a biological pathway. "

+ why is this about representation learning?
+ why adding uninformative unconnected points relates to representation learning?
+ why adding uninformative connected points relates to representation learning?
+ why and how the percolation task helps to predict 440 patients' survival from the gene expression profiles?

5. The novelty is limited. It mostly used Graph Convolution Net in a real-world task.

---

### Decision · Program_Chairs · 2018-03-20
**ICLR 2018 Workshop Acceptance Decision**

**Decision:**

Reject

**Comment:**

Based on the reviews, this paper has not been accepted for presentation at the ICLR workshop. However, the conversation and updates can continue to appear here on OpenReview.